# Quality of End-of-Life Care during the COVID-19 Pandemic at a Comprehensive Cancer Center

**DOI:** 10.3390/cancers15082201

**Published:** 2023-04-08

**Authors:** Yvonne Heung, Donna Zhukovsky, David Hui, Zhanni Lu, Clark Andersen, Eduardo Bruera

**Affiliations:** 1Department of Palliative Care, Rehabilitation and Integrative Medicine, The University of Texas MD Anderson Cancer Center, Houston, TX 77030, USA; yjheung@mdanderson.org (Y.H.);; 2Department of Biostatistics, The University of Texas MD Anderson Cancer Center, Houston, TX 77030, USA

**Keywords:** supportive care, palliative care, end-of-life care, advanced cancer, DNR, COVID-19

## Abstract

**Simple Summary:**

To better understand how the COVID-19 pandemic impacted end-of-life care in patients with advanced cancer at a comprehensive cancer center, we compared in-hospital deaths from April through July 2019 with those from April through July 2020. After the onset of the pandemic, do-not-resuscitate orders occurred earlier, palliative care referrals occurred earlier, fewer deaths occurred in the Intensive Care Unit, and more deaths occurred in the Palliative Care Unit. These outcomes suggest an improvement in the quality of end-of-life care in response to the COVID-19 pandemic. These encouraging findings may have future implications for maintaining quality end-of-life care post-pandemic.

**Abstract:**

To evaluate how the COVID-19 pandemic impacted the quality of end-of-life care for patients with advanced cancer, we compared a random sample of 250 inpatient deaths from 1 April 2019, to 31 July 2019, with 250 consecutive inpatient deaths from 1 April 2020, to 31 July 2020, at a comprehensive cancer center. Sociodemographic and clinical characteristics, the timing of palliative care referral, timing of do-not-resuscitate (DNR) orders, location of death, and pre-admission out-of-hospital DNR documentation were included. During the COVID-19 pandemic, DNR orders occurred earlier (2.9 vs. 1.7 days before death, *p* = 0.028), and palliative care referrals also occurred earlier (3.5 vs. 2.5 days before death, *p* = 0.041). During the pandemic, 36% of inpatient deaths occurred in the Intensive Care Unit (ICU) and 36% in the Palliative Care Unit, compared to 48 and 29%, respectively, before the pandemic (*p* = 0.001). Earlier DNR orders, earlier palliative care referrals, and fewer ICU deaths suggest an improvement in the quality of end-of-life care in response to the COVID-19 pandemic. These encouraging findings may have future implications for maintaining quality end-of-life care post-pandemic.

## 1. Introduction

The beginning of the novel coronavirus disease 2019 (COVID-19) pandemic challenged healthcare systems throughout the world to think critically about potential medical and ethical dilemmas, such as how to ration care in a resource-limited environment. Due to the high mortality associated with COVID-19, the involvement of palliative care and advance care planning discussions, such as those including values and preferences regarding cardiopulmonary resuscitation, became increasingly important at the onset of the pandemic. There were appropriate concerns that non-beneficial or unwanted cardiopulmonary resuscitation had the potential to place additional strain on essential healthcare workers, exhaust limited supplies of personal protective equipment, and increase the risk of psychological distress on patients and families [1,2].

Palliative care provided alongside disease-directed treatment has been associated with improved symptom burden, quality of life, prognostic awareness, patient and family satisfaction, decreased healthcare costs, location of death and improved survival [3,4,5,6,7]. In the intensive care unit setting, palliative care involvement has been associated with reduced length of stay, improved communication regarding goals of care, and greater family satisfaction [8,9]. In addition, patients at the end of life who receive referrals to palliative care are more likely to die at home or in an inpatient hospice facility [10,11]. The American Society of Clinical Oncology, National Comprehensive Cancer Network, European Society of Medical Oncology, and other national and international guidelines recommend timely referral to palliative care to optimize patient-centered and goal-concordant care as early as at the time of diagnosis through survivorship or the end-of-life [12,13,14]. Despite these recommendations, referrals to palliative care are commonly deferred until patients are profoundly symptomatic when treatment options become limited at the end of life or not considered at all [15,16,17,18,19,20].

In addition to providing complex symptom management, clarification of goals of care, and psychosocial and spiritual support throughout the cancer trajectory, palliative care involvement routinely facilitates value-based serious illness conversations. Ideally, discussions regarding end-of-life care preferences are incorporated into ongoing goals of care conversations and longitudinal treatment planning in the outpatient setting, where patients are more likely to have the capacity to clearly express their care preferences. Guidelines recommend initiating advance care planning for patients with serious or life-limiting illnesses in the months to years prior to death [21]; unfortunately, many discussions are often deferred until the end of life [18,22,23,24,25,26]. As a result, important conversations delayed until the time of clinical deterioration presents many challenges, such as unexpected shifts in decision-making responsibilities from patient to caregivers, increased caregiver posttraumatic symptoms [27], and missed opportunities to empower patients receiving palliative care. Delays in code status discussions have led to more aggressive end-of-life care, including medically non-beneficial interventions such as cardiopulmonary resuscitation in the setting of advanced cancer [28,29,30].

The urgency of the COVID-19 pandemic prompted clinicians to incorporate early and intentional conversations about values, goals, and preferences for care, especially for patients with serious illnesses at risk of clinical deterioration. These discussions were additionally complicated by social isolation related to strict visitation policies and the need to communicate remotely [31]. As healthcare systems begin to transition from crisis-oriented care toward more endemic models of care, it is necessary to understand how healthcare delivery has evolved in response to the COVID-19 pandemic to continue building on the lessons learned.

While an increasing number of studies have examined the involvement of palliative care and end-of-life outcomes, specifically in patients with COVID-19 [32,33,34,35,36], this study investigated the care received by patients with advanced cancer in response to the COVID-19 pandemic. The primary objective of this study was to compare the frequency and timing of do-not-resuscitate (DNR) orders during the terminal admission of patients with advanced cancer before and after the onset of the COVID-19 pandemic at a comprehensive cancer center. Secondarily, we compared sociodemographic and clinical characteristics, the timing of palliative care referral, the location of death, and out-of-hospital DNR documentation.

## 2. Materials and Methods

This retrospective study was approved by the Institutional Review Board at The University of Texas MD Anderson Cancer Center with a waiver of informed consent. We reviewed the electronic medical records of a random sample of 250 inpatient deaths from 1 April 2019 to 31 July 2019 (before the COVID-19 pandemic) as well as 250 consecutive inpatient deaths from 1 April 2020 to 31 July 2020 (after the onset of the COVID-19 pandemic) to compare the frequency and timing of DNR orders. We collected sociodemographic characteristics, including age, sex, race/ethnicity, and religion, as well as clinical characteristics, such as primary cancer diagnosis, the timing of palliative care referral, location of death, and completion of out-of-hospital DNR documentation prior to terminal admission.

As previously described [16], our inpatient palliative care consultation service operates seven days a week and consists of five teams staffed by palliative care–trained attending physicians, fellows, and advanced practice providers. Social workers, chaplains, and psychologists also make up the interdisciplinary team and become involved according to the needs of the patients and families. Upon receipt of the order for palliative care referral through the electronic medical record system, the consultation is immediately distributed to a team. Most referrals are seen within hours, with all consultations completed within 24 h. In anticipation of the COVID-19 pandemic, an additional palliative care team was created specifically to address the unique needs of patients in the Intensive Care Unit (ICU) and those who were at risk of clinical deterioration and required timely clarification of goals of care. All patients were tested for COVID-19 upon presentation to the hospital and at regular intervals during their admission. All patients who tested positive for COVID-19 received care in a separate and dedicated COVID-19 unit, where both non-invasive and invasive ventilatory support was available. Accordingly, the admission criteria or bed availability of the ICU was not impacted by the onset of the COVID-19 pandemic at our institution. Patients who were critically ill from acute or severe complications of their disease or treatment were considered for admission to the ICU.

Hakim et al. reported in their study a median of 3 days between DNR order and death, with an interquartile range from 1 to 7 days [37]. On the log(days+1) scale, the distances between the median and quartiles are symmetrical, and the standard deviation can be estimated as (log(7+1) − log(1+1))/1.35 = 1.027. With 250 patients per time point, a power analysis to estimate the percent change in the time between DNR and death between time points (pre-COVID versus post-COVID) based on a 2-sided 2-sample *t*-test with alpha = 0.05, a standard deviation of 1.027, and 80% power, ignoring left-censoring due to absence of DNR, was sufficient to detect differences between time points of as small as 0.258 on the log scale, which corresponds to a factor of exp(0.258) = 1.29, or a 29% change between time points.

Demographic and baseline clinical characteristics were summarized as mean and median with standard deviation or frequency with percentage, with differences between time points assessed by a 2-sided 2-sample I-test or chi-square test. Times from DNR order to death, from first palliative care referral to death, and from hospital admission to death were separately modeled with relation to COVID-19 time point using time-to-event methodologies. Kaplan-Meier methods were used to provide unadjusted summaries of median survival with 95% confidence intervals, as well as graphical summaries. To allow comparisons between time points while controlling for left censoring, accelerated failure time models were used to model time-to-event with relation to time point, with adjustment for demographic and baseline clinical covariates (age, sex, race/ethnicity, cancer type, location of death, and religion). For each model, Weibull, exponential, Gaussian, logistic, log-normal, and log-logistic distributions were considered, and the optimal distribution was selected on the basis of lower Akaike information criterion together with a good fit of a Kaplan-Meier plot of model residuals to the model distribution; models for the time from DNR order to death and from first palliative care referral to death utilized the Weibull distribution, and those for the time from hospital admission to death utilized the log-logistic distribution. Accelerated failure time models are beneficial because they provide intuitively understandable ratios of expected survival times between the time points, as opposed to less intuitive hazard ratios provided by Cox models.

Statistical analyses were performed using R statistical software, version 4.1.2 (R Core Team, 2021). In all statistical tests, the two-sided alpha was 0.05. Survival was modeled using the “survival” package. In the accelerated failure time models, the assessment of differences among discrete variable levels was estimated using the “emmeans” package, including adjusted means weighted proportionally to covariate marginal frequencies.

## 3. Results

A total of 500 inpatient deaths were included in this study. A random sample of 250 inpatient deaths was selected from before the COVID-19 pandemic (between 1 April 2019, to 31 July 2019) and compared to all 250 consecutive inpatient deaths from after the onset of the COVID-19 pandemic (1 April 2020 to 31 July 2020). Sociodemographic characteristics and primary diagnosis are summarized in Table 1. Overall, there were no significant differences in age, gender, race/ethnicity, or religion between the two cohorts. There was a significant difference in primary malignancy (*p* = 0.005), including a 10% decline in patients with leukemia after the onset of the COVID-19 pandemic.

The clinical outcomes are summarized in Table 2. There were no significant differences in pre-admission completion of out-of-hospital DNR documentation or the frequency of DNR orders between the two cohorts. However, after the onset of the COVID-19 pandemic, DNR orders occurred significantly earlier compared with before the COVID-19 pandemic [Kaplan-Meier median survival of 2.9 (95% confidence interval 2.4–3.9) days vs. 1.7 days (1.4–2.3), with a corresponding covariate-adjusted model survival time ratio of 1.47 (1.01–2.07), *p* = 0.028]. Figure 1 shows the time between DNR order and death increased between time points. Although not statistically significant, more patients after the onset of the COVID-19 pandemic received a referral to palliative care, and those referrals occurred significantly earlier during the terminal admission [3.5 (2.2–4.6) days vs. 2.5 (1.7–3.2) days prior to death, with covariate-adjusted survival time ratio of 1.35 (1.01–1.79), *p* = 0.041]. There was a significant difference in the location of death (*p* = 0.001), with 12% fewer deaths occurring in the ICU and 7% more deaths occurring in the palliative care unit after the onset of the COVID-19 pandemic. The overall length of stay showed a non-significant increasing trend after the onset of the COVID-19 pandemic [9.4 (8.6–10.8) days vs. 7 (6.3–8.6) days, with covariate-adjusted survival time ratio of 1.13 (0.93–1.38), *p* = 0.23].

## 4. Discussion

This study provides a unique examination of how the delivery of end-of-life care at a comprehensive cancer center has evolved in response to the COVID-19 pandemic. We report here several interesting observations related to the terminal admission of patients with advanced cancer after the onset of the COVID-19 pandemic, including earlier DNR orders, earlier referrals to palliative care, and fewer ICU deaths. These important findings indicate an improvement in the quality of end-of-life care.

Our finding that DNR orders for patients with advanced cancer occurred significantly earlier after the onset of the COVID-19 pandemic is especially encouraging as it highlights a positive shift in timely and intentional conversations between clinicians and patients, promoting optimal patient autonomy in end-of-life decision-making while capacity is more likely to be intact. As described in other studies, unfortunately, discussions around advance care planning and DNR orders frequently occur very late, within hours to days of death, for many different reasons [18,22,23,24,25,26]. While it can be argued that a peaceful and natural death is achieved as long as a DNR order has been placed before the moment of cardiopulmonary arrest, previous studies suggest that timing matters. Early DNR orders not only limit caregiver distress related to the burden of decision-making by substituted judgment and complicated grief during the bereavement process but also allow for discussions with patients around goal-concordant care during the terminal admission, such as avoiding invasive procedures and medically non-beneficial treatments prior to death [27,30,38,39]. A timely approach to appropriate discussions is necessary for reducing avoidable suffering and achieving quality end-of-life care.

This study provides evidence that DNR orders occurred earlier after the onset of the COVID-19 pandemic; however, we did not find a concomitant increase in the overall frequency of DNR orders. This is likely due to a ceiling effect as most decedents (at least 90%) both before and after the onset of the COVID-19 pandemic had a DNR order. A substantially larger sample size would have been required to detect a difference. The high frequency of DNR orders prior to death is consistent with existing studies that have analyzed DNR trends in patients with advanced cancer [23]. We also did not observe a concomitant increase in the frequency of pre-admission out-of-hospital DNR documentation, confirming findings by other studies that have observed this phenomenon during the COVID-19 pandemic [34]. Interestingly, despite a widespread sense of urgency to proactively shift advance care planning upstream toward the outpatient setting at the onset of the COVID-19 pandemic [1,40,41], out-of-hospital DNR documentation remained very low during the pandemic (5% completion before and 4% completion after the onset of the pandemic). While it is theorized that advance care planning in the outpatient setting facilitates inpatient goals of care discussions with the patient and surrogate decision-makers, it is unclear to what degree it ultimately ensures goal-concordant care and whether it is superior to other tools yet to be developed that can reliably serve as a valid quality indicator for end-of-life discussions and improve the quality of end-of-life care [42,43,44].

We identified that palliative care referrals occurred significantly earlier in the terminal admission after the onset of the COVID-19 pandemic. This is an important finding considering that patients with advanced cancer generally receive very late palliative care referrals (often within days of death) and exhibit a significant level of physical and emotional suffering upon presentation [15,16,17,18,19,20]. Furthermore, timely palliative care consultation allows for access to the palliative care unit and therefore avoids aggressive end-of-life care typically provided in the ICU. In this study, there was a significant decrease in ICU deaths and a concomitant increase in palliative care unit deaths, suggesting an improvement in the quality of end-of-life care after the onset of the COVID-19 pandemic [45,46,47,48]. Although not statistically significant, we are encouraged by the continued growth in palliative care referrals throughout the pandemic. Most patients who died in the inpatient setting at our comprehensive cancer center received a palliative care referral (60% before the COVID-19 pandemic and 68% after the onset of the COVID-19 pandemic). This sustained growth and high penetration rate are consistent with emerging literature describing similar observations for patients with COVID-19, underscoring the central role of palliative care for vulnerable populations during the pandemic and the unmet need for appropriate infrastructure to meet increasing demands for specialized palliative care [34,49,50]. Further research should focus on how to improve timely access to palliative care [51].

We report several findings related to operational changes unique to tertiary care comprehensive cancer centers that were implemented in anticipation of the potential surge at the beginning of the COVID-19 pandemic. Due to the nature of the specialized care provided at our cancer center, the characteristics of our patient population likely differ from those in the community setting. For example, the dramatic decline (10%) in inpatient deaths in patients with leukemia after the onset of the COVID-19 pandemic can be attributed to the suspension of interhospital transfers for select patients seeking specialized care. While most patients diagnosed with solid tumors are initially seen in consultation as an outpatient and generally considered for admission in the case of decompensation, many patients with hematologic malignancies initially present as an inpatient or require transfer to tertiary care centers for a higher level of care. Additionally, we observed a non-significant increase in length of stay after the onset of the COVID-19 pandemic, which may be attributed to an increase in patients with a need for ventilatory support as well as challenges in discharging patients to long-term acute care centers, skilled nursing facilities, or inpatient hospice units due to more restrictive acceptance criteria. More research is needed to further explain these results.

This study has several limitations. First, this study was conducted at a single tertiary care institution. This may limit the generalizability of our study as the patient demographics seeking care at our cancer center likely differ from those in the community. Only 5% of deaths occurred in our separate and dedicated COVID-19 unit that was designated for all patients who tested positive for COVID-19. Our comprehensive cancer center includes a palliative care unit that is staffed by a well-established and continually growing interdisciplinary palliative care team [52]. As mentioned in other studies, many palliative care teams remain understaffed during the COVID-19 pandemic, despite interdepartmental collaboration and support from outside institutions [34,36]. Therefore, similar gains may not be observed due to differences in structure and resources needed to deploy palliative care consultations in a timely manner. Second, the retrospective design of our study limited the capability to explore specific variables related to palliative care referrals, such as clinician characteristics, patient and family attitudes and beliefs, and details of individual advance care planning discussions. It is also difficult to infer causation for the outcomes described. Finally, this study focused on the outcomes of deceased patients before and after the onset of the COVID-19 pandemic. Future prospective studies may want to explore longer-term follow-up and capture a broader patient population, such as those discharged alive to home hospice or inpatient hospice units, to better understand how the delivery of healthcare for patients with advanced cancer has evolved in response to the COVID-19 pandemic.

## 5. Conclusions

During the COVID-19 pandemic, we observed significantly earlier referrals to palliative care, earlier DNR orders and fewer ICU deaths in patients with advanced cancer at our comprehensive cancer center. These findings indicate an improvement in the quality of end-of-life care and may have future implications for the timely integration of palliative care. Further research is needed to understand how to maintain and expand on such progress as healthcare systems transition toward an endemic model of care.

## Figures and Tables

**Figure 1 cancers-15-02201-f001:**
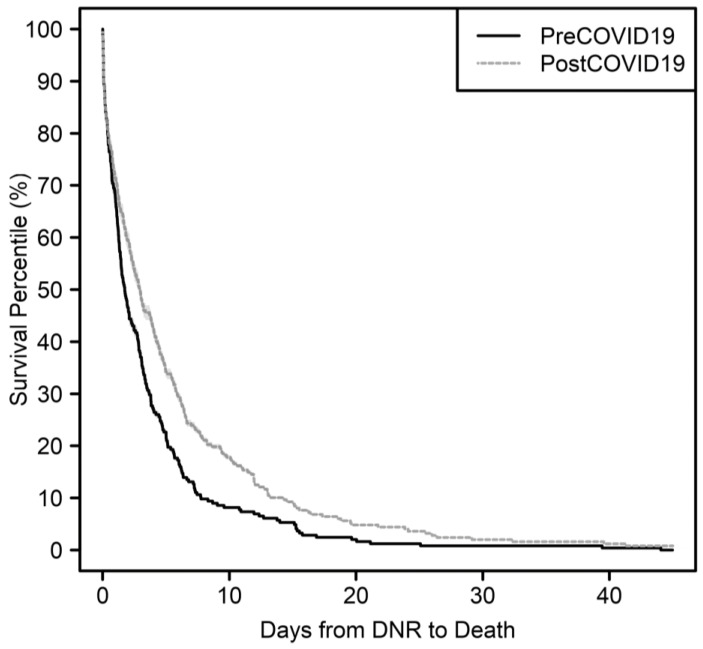
Kaplan-Meier summary of time from DNR order to death.

**Table 1 cancers-15-02201-t001:** Characteristics of inpatient deaths before and during the COVID-19 pandemic (n=500).

Patient Characteristics	Before COVID-19 (*n* = 250), n (%)	During COVID-19	*p* Value
		(n = 250), n (%)	
Age, years, median (IQR)	64 (54–72)	64 (54–73)	0.75
Gender			0.47
Female	116 (46)	107(43)	
Male	134 (54)	143 (57)	
Race/ethnicity			0.99
Asian	18 (7)	21 (8)	
Black or African American	42 (17)	38 (15)	
Caucasian	163 (65)	163 (65)	
Hispanic or Latino	23 (9)	24 (10)	
Other	4 (2)	4 (2)	
Religion			0.31
Catholic	52 (21)	62 (25)	
Christian	138 (55)	137 (55)	
Jewish	8 (3)	3 (1)	
Muslim	7 (3)	1 (0)	
None	16 (6)	20 (8)	
Other	8 (3)	9 (3)	
Unknown	21 (8)	19 (8)	
Cancer			0.005
Breast	19 (8)	13 (5)	
Gastrointestinal	41 (16)	34 (14)	
Genitourinary	10 (4)	31 (12)	
Gynecological	7 (3)	8 (3)	
Head & Neck	10 (4)	13 (5)	
Leukemia	80 (32)	56 (22)	
Lymphoma/myeloma	32 (13)	41 (16)	
Melanoma	4 (2)	1 (0)	
Neurological	2 (1)	4 (2)	
Other	3 (1)	10 (4)	
Sarcoma	11 (4)	8 (3)	
Thoracic	31 (12)	31 (12)	

IQR, interquartile range.

**Table 2 cancers-15-02201-t002:** Outcomes of inpatient deaths before and during the COVID-19 pandemic (n=500).

Patient Characteristics	Before COVID-19	During COVID-19	Before/During COVID-19	*p* Value
	n = 250	n = 250		
Out-of-hospital DNR, n (%)	13 (5)	10 (4)		0.67
Code status, n (%)				
DNR	226 (90)	234 (94)		0.25
Full code	24 (10)	16 (6)		
Location of death, n (%)				
Intensive Care Unit	120 (48)	89 (36)		0.001
Palliative Care Unit	73 (29)	91 (36)		
Medical/surgical floor	53 (21)	51 (20)		
COVID unit	0 (0)	13 (5)		
Other *	4 (2)	6 (2)		
Palliative care referral, n (%)	150 (60)	171 (68)		0.062
DNR prior to death, days, median survival (CI95)	1.7 (1.4–2.3)	2.9 (2.4–3.9)	STR (CI95): 1.57 (1.22–2.01)	0.028
PC prior to death, days, median survival (CI95)	2.5 (1.7–3.2)	3.5 (2.2–4.6)	STR (CI95): 1.35 (1.01–1.79)	0.041
Length of stay, days, median survival (CI95)	7 (6.3–8.6)	9.4 (8.6–10.8)	STR (CI95): 1.13 (0.93–1.38)	0.23

CI95, 95% confidence interval; DNR, do not resuscitate; PC, palliative care; STR: survival time ratio (approximately interpretable as the inverse of a hazard ratio) from covariate-adjusted accelerated failure time model, which adjusted for age, sex, race, cancer type, location of death, and religion. * Other units include the emergency room and operating room.

## Data Availability

Not applicable.

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
