# Peer review of "Quality of End-of-Life Care during the COVID-19 Pandemic at a Comprehensive Cancer Center"

_cancers, 2023, doi:10.3390/cancers15082201_

Round 1

Reviewer 1 Report

 Quality of End-of-Life Care during the COVID-19 Pandemic at a Comprehensive Cancer Center

The manuscript is on an important topic. I have, however, some major and a few minor comments.

MAJOR COMMENTS

The authors correctly comment (in the Limitations section) that this is a single-institution study. Even so, I am surprised to find that in their reference material from 2019, as many as 48% or their cancer patients die in intensive care units (ICU), which is an exceptionally high figure, as ICU should preferably not be the preferred place of death.

Moreover, the median age of death was 64 years both before and during the pandemic, which is lower than the expected median age of death for cancer patients.

Obviously, the patients at this institution constitute a selective group, that is not entirely representative of the underlying population. The authors should therefore a) describe their criteria for admission to the cancer hospital in the Introduction and b) comment on the representability and generalizability in the Discussion section.

The reference sample from 2019 is described as a random sample, but in what way is it “random”? Is it a consecutive sample or is any other sampling method applied (also considering that both groups are exactly 250 persons).

The study sample during the first wave of the pandemic: did all 250 patients die of cancer (only), or did some of them die of COVID-19? Were all patients tested?

If some of the study patients died of COVID-19, these should be described separately. If they died of COVID-19, it is surprising that the median age was only 64 years as COVID-deaths mainly affected elderly cancer patients (it is also surprising that the distribution between female and male was similar to 2019, as there was an overrepresentation of men dying of COVID-19).

If some of the patients died of COVID-19, how many of them required high-flow oxygen therapy or non-invasive or invasive respiratory support? If so – how did this impact on ICU care as such treatments are typically provided in ICUs?

Regarding the (very high) ICU figures 2019 and 2020: Where the indications for ICU admissions identical during 2019 and 2020? What were the main indications? As a reader, more details about the ICU care is needed in order to assess the relevance of the ICU care for this patient group of imminently dying cancer patients.

Table 1: There is a significant difference between the proportions of genitourinary cancers in 2019 and 2020 (10/250 i.e. 4% vs 31/250, i.e. 12%, which would give a p-value of .0006 in a simple 2x2 chi-2 calculation): were the significantly higher proportion in 2020 mainly persons with prostate cancer, i.e. male patients (which, in general, were over-represented as regards COViD-19 deaths )?

Although the DNR prior to death increased from 1.7 days to 2.9 days, both figures are extremely low if the study is about cancer deaths (not COVID-19 deaths). How is this explained? Clinically, it is in most cases obvious when the dying process has started, long before 1.7 days before death. If advance care planning (ACP) is timely performed, why is DNR performed not until a couple of days before the patient die? Is this a routine, typical for the institution or typical for the whole of the US? The authors should comment on these findings.

I have the same question as regards initiation of palliative care, which was done 2.5 and 3.5 days before death. In my clinical experience, this is very late. What is the main reason for this late referral? The authors should comment on these figures.

MINOR COMMENTS

The authors mention that health-care system begin to transition from crisis-oriented care toward more endemic models of care. I suggest that they describe what an endemic model of care is.

The study population is from the first wave of the COVID-pandemic. As the primary objective is to study changes in health care, in relation to the pandemic, why was the study limited to the first wave, as much of the changes today have been made later, when also vaccines are available?

The 250 deaths during April 1 to July 31: were these consecutive deaths or were they selected in some way (as they were exactly 250 – as in the comparison group)?

In all tests, the two-sided alpha was 0.05. Still, differences that did not reach the level are presented, e.g., in the Abstract: “

 Palliative care referrals increased (68% vs. 60%, p=0.062) ….”

Why?

Reviewer 2 Report

The study of the quality of EOL care for cancer patients in a single cancer center is interesting, especially when comparing before and after the COVID pandemic.

I have some concerns about the methodology and results of this study.

1. As a researcher in palliative care, a median of fewer than 3 days between DNR orders and deaths is quite disappointing because it indicates a delayed intervention of palliative care for cancer patients. Is it a phenomenon in this cancer center? Or is it common for most settings with cancer treatments?

2. The authors deduced that death in the PC unit meant better palliative care than that in the ICU. However, we know that the resource of the ICU is limited during the COVID pandemic. How can your control this confounding before deducing the statement of improved palliative care during the pandemic?

3. In the results, the authors do not explain the Kaplan-Meier curve. It is only a short statement "as shown in Figure 1". The "2.9 days vs. 1.7 days, p=0.024" is a result of Table 2. Please describe the findings and statistical values of the Kaplan-Meier analysis.

4. Table 2 depicts the outcomes of interest in this study, but it is not adjusted for different demographics between 2 groups as shown in Table 1. Age, gender and cancer type may need adjusted to allow comparison between two groups.

Reviewer 3 Report

I would like to thank the authors and the Editorial Board for the opportunity to review the article submitted to MDPI’s Cancers. The authors' manuscript refers to a very important topic: the care quality of oncological patients during the COVID-19 pandemic. The presented manuscript presents the results based on one’s institution data, but despite that, provides important practical information about the care quality during the pandemic. I believe that the presented article is of high quality, but some minor changes could still improve it. Below I present my comments on the individual sections of the mentioned manuscript.

Methods: Here authors present the results of the sensitivity power analysis. It is unknown why the authors took the power threshold value of 1-Beta=0.80 instead of 0.95. Please elaborate.

Results: Please supplement Tables 1 and 2 with effect size measures. Frequencies between subgroups can be represented by Yule’s Phi (for 2x2 setting) or Cramer’s V (for 2x3+ setting), and between groups, t-test analyses require to report Cohen’s d values.

Results: What is more, the authors did not report very important test statistics (such as t-values or chi-squared values) or degrees of freedom, therefore it is impossible to verify if the authors’ results are free of any p-hacking practices. I highly recommend that the authors supplement that data.

Results and Discussion: The authors describe their results based on the p-value levels. P-values are highly correlated with the size of the studied sample (see Lakens, 2022). I highly recommend that authors present and discuss their results based on the size of the effect rather than its significance levels.

Round 2

Reviewer 1 Report

The manuscript is now improved and I have no further comments.